# Multi-Stage Influence Function

**Hongge Chen**[*,1]    **Si Si**[2]    **Yang Li** [2]    **Ciprian Chelba** [2]
**Sanjiv Kumar** [2]    **Duane Boning** [1]    **Cho-Jui Hsieh** [3]
[1]MIT    [2] Google Research    [3]UCLA
[*]Work was done when interning at Google Research.
chenhg@mit.edu  sisidaisy@google.com  liyang@google.com  ciprianchelba@google.com
sanjivk@google.com  boning@mtl.mit.edu  chohsieh@cs.ucla.edu

## Abstract

Multi-stage training and knowledge transfer, from a large-scale pretraining task to various finetuning tasks, have revolutionized natural language processing and computer vision resulting in state-of-the-art performance improvements. In this paper, we develop a multi-stage influence function score to track predictions from a finetuned model all the way back to the pretraining data. With this score, we can identify the pretraining examples in the pretraining task that contribute most to a prediction in the finetuning task. The proposed multi-stage influence function generalizes the original influence function for a single model in (Koh & Liang, 2017), thereby enabling influence computation through both pretrained and finetuned models. We study two different scenarios with the pretrained embeddings fixed or updated in the finetuning tasks. We test our proposed method in various experiments to show its effectiveness and potential applications.

## 1   Introduction

Multi-stage training has become increasingly important and has achieved state-of-the-art results in many tasks. In natural language processing (NLP) applications, it is now a common practice to first learn word embeddings (e.g., word2vec [16], GloVe [19]) or contextual representations (e.g., ELMo [20], BERT [7]) from a large unsupervised corpus, and then refine or finetune the model on supervised end tasks. Also, transfer learning has been widely used in many different tasks. Intuitively, the successes of these multi-stage learning paradigms are due to knowledge transfer from pretraining tasks to the end task. However, current approaches using multi-stage learning are usually based on trial-and-error and many fundamental questions remain unanswered. For example, which part of the pretraining data/task contributes most to the end task? How can one detect "false transfer" where some pretraining data/task could be harmful for the end task? If a testing point is wrongly predicted by the finetuned model, **can we trace back to the problematic examples in the pretraining data?** Answering these questions requires a quantitative measurement of how the data and loss function in the pretraining stage influence the end model, which has not been studied in the past and is the main focus of this paper.

To find the most influential training data responsible for a model's prediction, the influence function was first introduced by [5]. Recently, as large-scale applications become more challenging for influence function computation, [13] proposed to use a first-order approximation to measure the effect of removing one training point on the model's prediction, to overcome computational challenges. These methods are widely used in model debugging and there are also some applications in machine learning fairness [2, 26]. However, all of the existing influence score computation algorithms study the case of single-stage training – where there is only one model with one set of training/prediction data in the training process. **To the best of our knowledge, the influence of pretraining data on a subsequent finetuning task and model has not been studied,** and it is nontrivial to apply the

original influence function in [13] to this scenario. A naive approach to solve this problem is to remove each individual instance out of the pretraining data one at a time and retrain both pretrain and finetune models; this is prohibitively expensive, especially given that pretraining models are often large-scale and may take days to train.

In this work, we study the influence function from pretraining data to the end task, and propose a novel approach to estimate the influence scores in multi-stage training that requires no additional retrain, does not require model convexity, and is computationally tractable. The proposed approach is based on the definition of influence function, and considers estimating influence score under two multi-stage training settings depending on whether the embedding from pretraining model is retrained in the finetuning task. The derived influence function well explains how pretraining data benefits the finetuning task. In summary, our contributions are threefold:

1. **We propose a novel estimation of influence score for multi-stage training.** In real datasets and experiments across various tasks, our predicted and actual influence score of the pretraining data to the finetuned model are well correlated. This shows the effectiveness of our proposed technique for estimating influence scores in multi-stage models.

2. **We propose effective methods to determine how testing data from the finetuning task is impacted by changes in the pretraining data.** We show that the influence of the pretraining data to the finetuned model consists of two parts: the influence of the pretraining data on the pretrained model, and influence of the pretraining data on the finetuned model. The impact can be quantified using our proposed technique.

3. **We propose methods to decide whether the pretraining data can benefit the finetuning task.** We show that the influence of the pretraining data on the finetuning task is highly dependent on 1) the similarity of two tasks or stages, and 2) the number of training data in the finetuning task. Our proposed technique provides a novel way to measure how the pretraining data helps or benefits the finetuning task.

## 2 Related Work

Multi-stage model training that trains models in many stages on different tasks to improve the end-task has been used widely in many machine learning areas. For example, transfer learning has been widely used to transfer knowledge from source task to the target task [18]. More recently, researchers have shown that training the computer vision or NLP encoder on a source task with huge amount of data can often benefit the performance of small end-tasks, and these techniques including BERT [7], Elmo [14] and large ResNet pretraining [15] have achieved state-of-the-arts on many tasks.

Although mutli-stage models have been widely used, there are few works on understanding multi-stage models and exploiting the influence of the training data in the pretraining step to benefit the fine-tune task. In contrast, there are many works that focus on understanding single stage machine learning models and explaining model predictions. Algorithms developed along this line of research can be categorized into features based and data based approaches. Feature based approaches aim to explain predictions with respect to model variables, and trace back the contribution of variables to the prediction [17, 9, 21, 23, 22, 25, 8, 6, 1]. However, they are not aiming for attributing the prediction back to the training data.

On the other hand, data based approaches seek to connect model prediction and training data, and trace back the most influential training data that are most responsible for the model prediction. Among them, the influence function [5, 13], which aims to model the prediction changes when training data is added/removed, has been shown to be effective in many applications. There is a series of work on influence functions, including investigating the influence of a group of data on the prediction [12], using influence functions to detect bias in word embeddings [2], and using it in preventing data poisoning attacks [24]. There are also works on data importance estimation to explain the model from the data perspective [27, 10, 11].

All of these previous works only consider a single stage training procedure, and it is not straightforward to apply them to multi-stage models. In this paper, we propose to analyze the influence of pretraining data on predictions in the subsequent finetuned model and end task.

# 3   Algorithms

In this section, we detail the procedure of multi-stage training, show how to compute the influence score for the multi-stage training, and then discuss how to scale up the computation.

## 3.1   Multi-Stage Model Training

Multi-stage models, which train different models in consecutive stages, have been widely used in various ML tasks. Mathematically, let $\mathcal{Z}$ be the training set for pretraining task with data size $|\mathcal{Z}| = m$, and $\mathcal{X}$ be the training data for the finetuning task with data size $|\mathcal{X}| = n$. In pretraining stage, we assume the parameters of the pretrained network have two parts: the parameters $W$ that are shared with the end task, and the task-specific parameters $U$ that will only be used in the pretraining stage. Note that $W$ could be a word embedding matrix (e.g., in word2vec) or a representation extraction network (e.g., Elmo, BERT, ResNet), while $U$ is usually the last few layers that corresponds to the pretraining task. After training on the pretraining task, we obtain the optimal parameters $W^*, U^*$. The pretraining stage can be formulated as

$$W^*,\ U^* = \arg\min_{W,U} \frac{1}{m} \sum_{z \in \mathcal{Z}} g(z,\ W,\ U) := \arg\min_{W,U} G(W,U), \tag{1}$$

where $g(\cdot)$ is the loss function for the pretrain task and $G(\cdot)$ is summation of loss with respect to all the pretraining examples. In the finetuning stage, the network parameters are $W, \Theta$, where $W$

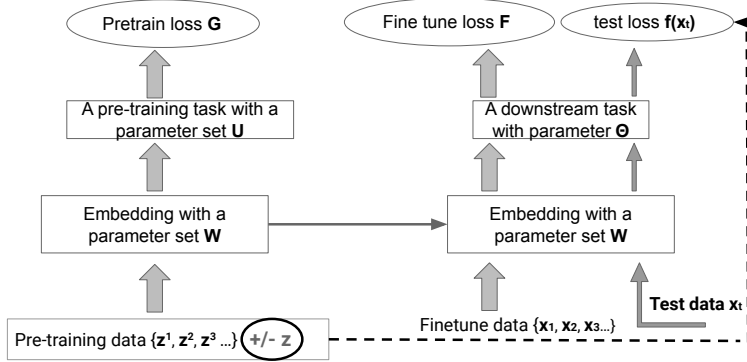

Figure 1: The setting for influence functions in multi-stage models. We consider a two-stage model, where we have a pretrained model, and a finetuned model for a desired end task. We seek to compute the influence of the pretraining data on predictions using testing data in the finetuning task.

is shared with the pretraining task and $\Theta$ is the rest of the parameters specifically associated with the finetuning task. We will initialize the $W$ part by $W^*$. Let $f(\cdot)$ denote the finetuning loss, and $F(\cdot)$ summarizes all the loss with respect to finetuning data, there are two cases when finetuning the end-task:

- Finetuning Case 1: Fixing embedding parameters $W = W^*$, and only finetune $\Theta$:

$$\Theta^* = \arg\min_{\Theta} \frac{1}{n} \sum_{x \in \mathcal{X}} f(x,\ W^*,\ \Theta) := \arg\min_{\Theta} F(W^*, \Theta). \tag{2}$$

- Finetuning Case 2: finetune both the embedding parameters $W$ (initialized from $W^*$) and $\Theta$. Sometimes updating the embedding parameters $W$ in the finetuning stage is necessary, as the embedding parameters from the pretrained model may not be good enough for the finetuning task. This corresponds to the following formulation:

$$W^{**}, \Theta^* = \arg\min_{W,\Theta} \frac{1}{n} \sum_{x \in \mathcal{X}} f(x,\ W,\ \Theta) := \arg\min_{W,\Theta} F(W, \Theta). \tag{3}$$

## 3.2 Influence function for multi-stage models

We derive the influence function for the multi-stage model to trace the influence of pretraining data on the finetuned model. In Figure 1 we show a flow chart of computing multi-stage influence function, the task we are interested in solving in this paper. Note that we use the same definition of influence function as [13] and discuss how to compute it in the multi-stage training scenario. As discussed at the end of Section 3.1, depending on whether or not we are updating the shared parameters $W$ in the finetuning stage, we will derive the influence functions under two different scenarios.

### 3.2.1 Case 1: embedding parameters $W$ are fixed in finetuning

To compute the influence of pretraining data on the finetuning task, the main idea is to perturb one data example in the pretraining data, and study how that impacts the test data. Mathematically, if we perturb a pretraining data example $z$ with loss change by a small $\epsilon$, the perturbed model can be defined as

$$\hat{W}_\epsilon, \hat{U}_\epsilon = \arg\min_{W,U} G(W, U) + \epsilon g(z, W, U). \tag{4}$$

Note that choices of $\epsilon$ can result in different effects in the loss function from the original solution in (1). For instance, setting $\epsilon = -\frac{1}{m}$ is equivalent to removing the sample $z$ in the pretraining dataset.

For the finetuning stage, since we consider Case 1 where the embedding parameters $W$ are fixed in the finetuning stage, the new model for the end-task or finetuning task will thus be

$$\hat{\Theta}_\epsilon = \arg\min_{\Theta} F(\hat{W}_\epsilon, \Theta). \tag{5}$$

The influence function that measures the impact of a small $\epsilon$ perturbation on $z$ to the finetuning loss on a test sample $x_t$ from finetuning task is defined as

$$I_{z,x_t} := \frac{\partial f(x_t, \hat{W}_\epsilon, \hat{\Theta}_\epsilon)}{\partial \epsilon}\Big|_{\epsilon=0} = \nabla_\Theta f(x_t, W^*, \Theta^*)^T \cdot I_{z,\Theta} + \nabla_W f(x_t, W^*, \Theta^*)^T \cdot I_{z,W}, \tag{6}$$

$$with \ I_{z,\Theta} := \frac{\partial \hat{\Theta}_\epsilon}{\partial \epsilon}\Big|_{\epsilon=0} \ and \ I_{z,W} := \frac{\partial \hat{W}_\epsilon}{\partial \epsilon}\Big|_{\epsilon=0} , \tag{7}$$

where $I_{z,\Theta}$ measures the influence of $z$ on the finetuning task parameters $\Theta$, and $I_{z,W}$ measures how $z$ influences the pretrained model $W$. Therefore we can split the influence of $z$ on the test sample into two pieces: one is the impact of $z$ on the pretrained model $I_{z,W}$, and the other is the impact of $z$ on the finetuned model $I_{z,\Theta}$. It is worth mentioning that, due to linearity, if we want to estimate a set of test example influence function scores with respect to a set of pretraining examples, we can simply sum up the pair-wise influence functions, and so define

$$I_{\{z^{(i)}\},\{x_t^{(j)}\}} := \sum_i \sum_j I_{z^{(i)},x_t^{(j)}}, \tag{8}$$

where $\{z^{(i)}\}$ contains a set of pretraining data and $\{x_t^{(j)}\}$ contains a group of finetuning test data that we are targeting on. Next we will derive these two influence scores $I_{z,\Theta}$ and $I_{z,W}$ (see the detailed derivations in the appendix) in Theorem 1 below.

**Theorem 1.** *For the two-stage training procedure in* (1) *and* (2)*, we have*

$$I_{z,W} := \frac{\partial \hat{W}_\epsilon}{\partial \epsilon}\Big|_{\epsilon=0} = -\left[\left(\frac{\partial^2 G(W^*, U^*)}{\partial(W, U)^2}\right)^{-1}\left(\frac{\partial g(z, W^*, U^*)}{\partial(W, U)}\right)\right]_W \tag{9}$$

$$I_{z,\Theta} := \frac{\partial \hat{\Theta}_\epsilon}{\partial \epsilon}\Big|_{\epsilon=0} = \left(\frac{\partial^2 F(W^*, \Theta^*)}{\partial \Theta^2}\right)^{-1} \cdot \left(\frac{\partial^2 F(W^*, \Theta^*)}{\partial \Theta \partial W}\right) \cdot \left[\left(\frac{\partial^2 G(W^*, U^*)}{\partial(W, U)^2}\right)^{-1}\left(\frac{\partial g(z, W^*, U^*)}{\partial(W, U)}\right)\right]_W \tag{10}$$

*where* $[\cdot]_W$ *means taking the $W$ part of the vector.*

By plugging (9) and (10) into (6), we finally obtain the influence score of pretraining data $z$ on the finetuning task testing point $x_t$, $I_{z,x_t}$ as

$$I_{z,x_t} = \left[-\frac{\partial f(x, W^*, \Theta^*)^T}{\partial \Theta} \cdot \left(\frac{\partial^2 F(W^*, \Theta^*)}{\partial \Theta^2}\right)^{-1} \cdot \frac{\partial^2 F(W^*, \Theta^*)}{\partial \Theta \partial W} + \frac{\partial f(x, W^*, \Theta^*)^T}{\partial W}\right] I_{z,W} \tag{11}$$

The pseudocode for the influence function in (11) is shown in Algorithm 1.

**Algorithm 1:** Multi-Stage Influence Score with Fixed Embedding

---

**Input:** pretrain and finetune models with $W^*$, $\Theta^*$, and $U^*$; pretrain and finetune training data $\mathcal{Z}$ and $\mathcal{X}$; test example $x_t$; and a pretrain training example $z$.

**Output:** Influence function value $I_{z,x_t}$.

1 Compute fintune model's gradients $\frac{\partial f(x_t, W^*, \Theta^*)}{\partial \Theta}$ and $\frac{\partial f(x_t, W^*, \Theta^*)}{\partial W}$;

2 Compute the first inverse Hessian vector product $V_{ihvp1}(x_t) := (\frac{\partial^2 F(W^*, \Theta^*)}{\partial \Theta^2})^{-1} \frac{\partial f(x_t, W^*, \Theta^*)}{\partial \Theta}$;

3 Compute finetune loss's gradient w.r.t $W$: $\frac{\partial f(x_t, W^*, \Theta^*)^T}{\partial W} = V_{ihvp1}^T \frac{\partial^2 F(W^* \Theta^*)}{\partial \Theta \partial W} - \frac{f(x_t, W^*, \Theta^*)}{\partial W}$
  and concatenate it with 0 to make it the same dimension as $(W, U)$;

4 Compute and save the second inverse Hessian vector product
  $V_{ihvp2}^T(x_t) := [\frac{\partial f(x_t, \Theta^*, W^*)^T}{\partial W}, \; 0](\frac{\partial^2 G(W^*, U^*)}{\partial(W,U)^2})^{-1}$ ;

5 Compute influence function score $I_{z,x_t} = V_{ihvp2}^T(x_t)\frac{\partial g(z, W^*, U^*)}{\partial(W,U)}$;

---

### 3.2.2 Case 2: embedding parameter $W$ is also updated in the finetuning stage

For the second finetuning stage case in (3), we will also further train the embedding parameter $W$ from the pretraining stage. When $W$ is also updated in the finetuning stage, it is challenging to characterize the influence since the pretrained embedding $W^*$ is only used as an initialization. In general, the final model $(W^{**}, \Theta^*)$ may be totally unrelated to $W^*$; for instance, when the objective function is strongly convex, any initialization of $W$ in (3) will converge to the same solution.

However, in practice the initialization of $W$ will strongly influence the finetuning stage in deep learning, since the finetuning objective is usually highly non-convex and initializing $W$ with $W^*$ will converge to a local minimum near $W^*$. Therefore, we propose to approximate the whole training procedure as

$$\bar{W}, \bar{U} = \underset{W,U}{\arg\min}\, G(W, U) \tag{12}$$

$$W^*, \Theta^* = \underset{W,\Theta}{\arg\min}\{\alpha\|W - \bar{W}\|_F^2 + F(W, \Theta)\},$$

where $\bar{W}, \bar{U}$ are optimal for the pretraining stage, $W^*, \Theta^*$ are optimal for the finetuning stage, and $0 \le \alpha \ll 1$ is a small value. This is to characterize that in the finetuning stage, we are targeting to find a solution that minimizes $F(W, \Theta)$ and is close to $\bar{W}$. In this way, the pretrained parameters are connected with finetuning task and thus influence of pretraining data to the finetuning task can be tractable. The results in our experiments show that with this approximation, the computed influence score can still reflect the real influence quite well.

Similarly we can have $\frac{\partial \hat{\Theta}_\epsilon}{\partial \epsilon}$, $\frac{\partial \hat{W}_\epsilon}{\partial \epsilon}$, and $\frac{\partial \bar{W}_\epsilon}{\partial \epsilon}$ to measure the difference between their original optimal solutions in (12) and the optimal solutions from $\epsilon$ perturbation over the pretraining data $z$. Similar to (6), the influence function $I_{z,x_t}$ that measures the influence of $\epsilon$ perturbation to pretraining data $z$ on test sample $x_t$'s loss is

$$I_{z,x_t} := \frac{\partial f(x_t, \hat{W}_\epsilon, \hat{\Theta}_\epsilon)}{\partial \epsilon}\Big|_{\epsilon=0} = \frac{\partial f(x_t, W^*, \Theta^*)}{\partial(W, \Theta)}^T \begin{bmatrix} \frac{\partial \hat{W}_\epsilon}{\partial \epsilon}\big|_{\epsilon=0} \\ \frac{\partial \hat{\Theta}_\epsilon}{\partial \epsilon}\big|_{\epsilon=0} \end{bmatrix}. \tag{13}$$

The influence function of small perturbation of $G(W, U)$ to $\bar{W}, W^*, \Theta^*$ can be computed following the same approach in Subsection 3.2.1 by replacing $\bar{W}$ for $W^*$ and $[\Theta^*, W^*]$ for $\Theta^*$ in (9). This will lead to

$$\frac{\partial \bar{W}_\epsilon}{\partial \epsilon}\Big|_{\epsilon=0} = -\left[(\frac{\partial^2 G(\bar{W}, \bar{U})}{\partial(W,U)^2})^{-1}(\frac{\partial g(z, \bar{W}, \bar{U})}{\partial(W,U)})\right]_W, \tag{14}$$

$$\begin{bmatrix} \frac{\partial \hat{\Theta}_\epsilon}{\partial \epsilon}\big|_{\epsilon=0} \\ \frac{\partial \hat{W}_\epsilon}{\partial \epsilon}\big|_{\epsilon=0} \end{bmatrix} = \begin{bmatrix} \frac{\partial^2 F(W^*, \Theta^*)}{\partial \Theta^2} & \frac{\partial^2 F(W^*, \Theta^*)}{\partial \Theta \partial W} \\ \frac{\partial^2 F(W^*, \Theta^*)}{\partial \Theta \partial W} & \frac{\partial^2 F(W^*, \Theta^*)}{\partial W^2} + 2\alpha I \end{bmatrix}^{-1} \cdot \begin{bmatrix} 0 \\ -2\alpha I \end{bmatrix} \cdot \left[(\frac{\partial^2 G(\bar{W}, \bar{U})}{\partial(W,U)^2})^{-1}(\frac{\partial g(z, \bar{W}, \bar{U})}{\partial(W,U)})\right]_W. \tag{15}$$

After plugging (14) and (15) into (13), we will have the influence function $I_{z,x_t}$.

Similarly, the algorithm for computing $I_{z,x_t}$ for Case 2 can follow Algorithm 1 for Case 1 by replacing gradient computation. Through the derivation we can see that our proposed multi-stage influence function does not require model convexity.

### 3.3 Computation Challenges

The influence function computation for multi-stage model is presented in the previous section. As we can see in Algorithm 1 that the influence score computation involves many Hessian matrix operations, which will be very expensive and sometimes unstable for large-scale models. We used several strategies to speed up the computation and make the scores more stable.

**Large Hessian Matrices**  As we can see from Algorithm 1, our algorithm involves several Hessian inverse operations, which is known to be computation and memory demanding. For a Hessian matrix $H$ with a size of $p \times p$ and $p$ is the number of parameters in the model, it requires $p \times p$ memory to store $H$ and $O(p^3)$ operations to invert it. Therefore, for large deep learning models with thousands or even millions of parameters, it is almost impossible to perform Hessian matrix inverse. Similar to [13], we avoid explicitly computing and storing the Hessian matrix and its inverse, and instead compute product of the inverse Hessian with a vector directly. More specifically, every time when we need an inverse Hessian vector product $v = H^{-1}b$, we invoke conjugate gradients (CG), which transforms the linear system problem into an quadratic optimization problem $H^{-1}b \equiv \arg\min_x\{\frac{1}{2}x^T H x - b^T x\}$. In each iteration of CG, instead of computing $H^{-1}b$ directly, we will compute a Hessian vector product, which can be efficiently done by backprop through the model twice with $O(p)$ time complexity [4].

The aforementioned conjugate gradient method requires the Hessian matrix to be positive definite. However, in practice the Hessian may have negative eigenvalues, since we run a SGD and the final Hessian matrix $H$ may not at a local minimum exactly. To tackle this issue, we solve

$$\arg\min_x\{\frac{1}{2}x^T H^2 x - b^T H x\}, \tag{16}$$

whose solution can be shown to be the same as $\arg\min_x\{\frac{1}{2}x^T H x - b^T x\}$ since the Hessian matrix is symmetric. $H^2$ is guaranteed to be positive definite as long as $H$ is invertible, even when $H$ has negative eigenvalues. If $H^2$ is not ill-conditioned, we can solve (16) directly. The rate of convergence of CG depends on $\frac{\sqrt{\kappa(H^2)}-1}{\sqrt{\kappa(H^2)}+1}$, where $\kappa(H^2)$ is the condition number of $H^2$, which can be very large if $H^2$ is ill-conditioned. When $H^2$ is ill-conditioned, to stabilize the solution and to encourage faster convergence, we add a small damping term $\lambda$ on the diagonal and solve $\arg\min_x\{\frac{1}{2}x^T(H^2 + \lambda I)x - b^T H x\}$.

**Time Complexity**  As mentioned above, we can get an inverse Hessian vector product in $O(p)$ time if the Hessian is with size $p \times p$. To analyze the time complexity of Algorithm 1, assume there are $p_1$ parameters in our pretrained model and $p_2$ parameters in our finetuned model, it takes $O(mp_1)$ or $O(np_2)$ to compute a Hessian vector product, where $m$ is the number of pretraining examples and $n$ is the number of finetuning examples. For the two inverse Hessian vector products as shown in Algorithm 1, the time complexity therefore is $O(np_2 r)$ and $O(mp_1 r)$, where $r$ is the number of iterations in CG. For other operations in Algorithm 1, vector product has a time complexity of $O(p_1)$ or $O(p_2)$, and computing the gradients of all pretraining examples has a complexity of $O(mp_1)$. So the total time complexity of computing a multi-stage influence score is $O((mp_1 + np_2)r)$. Therefore we can see that the computation is tractable as it is linear to the number of training samples and model parameters. All the computation related to inverse Hessian can use inverse Hessian vector produc (IHVP), which makes the memory usage and computation efficient.

## 4 Experiments

In this section, we will conduct experiment on real datasets in both vision and NLP tasks to show the effectiveness of our proposed method. Our code will be available in the Github Repository of Google Research.

### 4.1 Evaluation of Influence Score Estimation

We first evaluate the effectiveness of our proposed approach for the estimation of influence function. For this purpose, we build two CNN models based on CIFAR-10 and MNIST datasets. The model structures are shown in Table A in Appendix. For both MNIST and CIFAR-10 models, CNN layers are used as embeddings and fully connected layers are task-specific. At the pretraining stage, we train

the models with examples from two classes ("bird" vs. "frog") for CIFAR-10 and four classes (0, 1, 2, and 3) for MNIST. The resulting embedding is used in the finetuning tasks, where we finetune the model with the examples from the remaining eight classes in CIFAR-10 or the other 6 numbers in MNIST for classification task.

We test the correlation between individual pretraining example's multi-stage influence function and the real loss difference when the pretraining examples are removed. We test two cases (as mentioned in Section 3.1) – where the pretrained embedding is fixed, and where it is updated during finetuning. For a given example in the pretraining data, we calculate its influence function score with respect to each test example in the finetuning task test set using the method presented in Section 3. To evaluate this pretraining example's contribution to the overall performance of the model, we sum up the influence function scores across the whole test set in the finetuning task.

To validate the score, we remove that pretraining example and go through the aforementioned process again by updating the model. Then we run a linear regression between the true loss difference values obtained and the influence score computed to show their correlation. The detailed hyperparameters used in these experiments are presented in Appendix B.

**Embedding is fixed** Figure 2(a) and Figure A in the appendix demonstrate the correlation results of CIFAR-10 and MNIST models when the embedding is fixed in finetuning task training. Similar to [13], we report study 100 training examples with the largest absolute influence function score values. From Figure 2(a) and Figure A we can see that there is a linear correlation between the true loss difference and the influence function scores obtained. The correlation is evaluated with Pearson's $r$ value. It is almost impossible to get the exact linear correlation because the influence function is based on the first-order conditions (gradients equal to zero) of the loss function, which may not hold in practice. The results in [13] show that their $r$ value is around 0.8 but their correlation is based on a single model with a single data source, but we consider a much more complex case with two models and two data sources: the relationship between pretraining data and finetuning loss function. So we expect to have a lower $r$ value. Therefore 0.6 is reasonable to show a strong correlation between pretraining data's influence score and finetuning loss difference. This supports our argument that we can use this score to detect the examples in the pretraining set which contributes most to the model's performance.

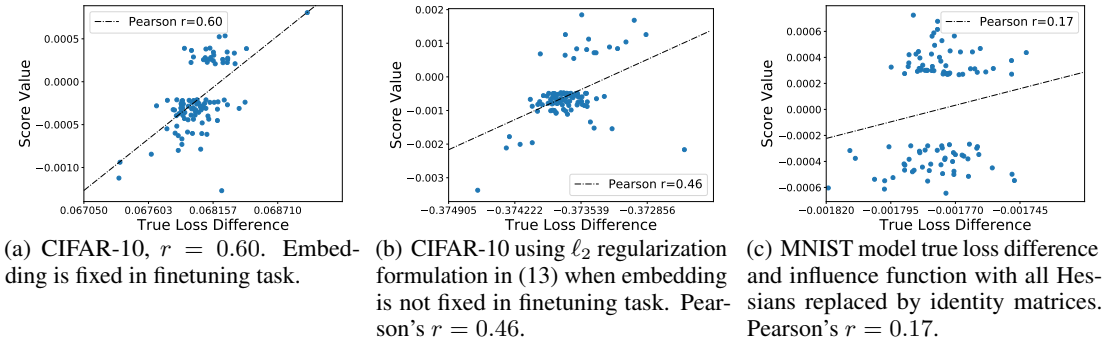

(a) CIFAR-10, $r = 0.60$. Embedding is fixed in finetuning task.

(b) CIFAR-10 using $\ell_2$ regularization formulation in (13) when embedding is not fixed in finetuning task. Pearson's $r = 0.46$.

(c) MNIST model true loss difference and influence function with all Hessians replaced by identity matrices. Pearson's $r = 0.17$.

Figure 2: True loss difference vs. the influence function scores by our proposed method.

One may doubt the effectiveness of the expensive inverse Hessian computation in our formulation. As a comparison, we replace all inverse Hessians in (11) with identity matrices to compute the influence function score for the MNIST model. The results are shown in Figure 2(c) with a much smaller Pearson's $r$ of 0.17. This again shows effectiveness of our proposed influence function.

**Embedding is updated in finetune** Practically, the embedding is also updated in finetuning. In Figure 2(b) we show the correlation between true loss difference and influence function score values using (12). We can see that even under this challenging condition, our multi-stage influence function from (12) still has a strong correlation with the true loss difference, with a Pearson's $r = 0.46$.

In Figure 3 we demonstrate the misclassified test images in the finetuning task and their corresponding largest positive influence score (meaning most influential) images in the pretraining dataset. Examples with large positive influence score are expected to have negative effect on the model's performance

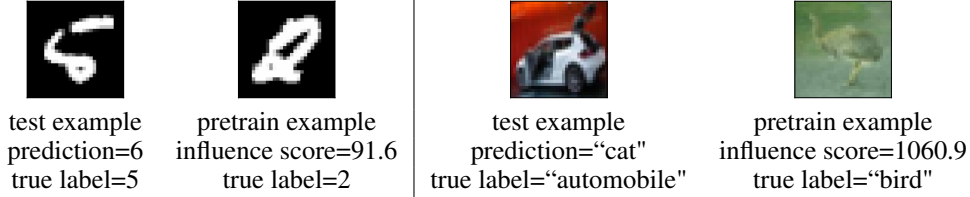

| test example<br>prediction=6<br>true label=5 | pretrain example<br>influence score=91.6<br>true label=2 | test example<br>prediction="cat"<br>true label="automobile" | pretrain example<br>influence score=1060.9<br>true label="bird" |

Figure 3: Identifying the pretrain example with largest influence function score which contributes an error in the finetune task. In this figure, we pair a misclassified test image in the finetuning task with the pretraining image which has the largest positive influence score value with respect to that test image. Intuitively, the identified pretraining image contributes most to the test image's error. We can indeed see that the identified examples are of low quality, which leads to negative transfer.

since intuitively when they are added to the pretraining dataset, the loss of the test example will increase. From Figure 3 we can indeed see that the identified examples are with low quality, and they can be easily misclassified even with human eyes.

## 4.2 Data Cleansing using Predicted Influence Score

Since the pretraining examples with large positive influences scores are the ones that will increase the loss function value indicating negative transfer. Based on the influence score computed, we can improve the negative transfer issue. We perform experiment on the CIFAR-10 dataset with the same setting as Section 4.1. After we removed the top 10% highest influence scores (positive values) examples from pretrain (source data), we can improve the accuracy on target data from 58.15% to 58.36%. As a reference, if we randomly remove 10% of pretraining data, the accuracy will drop to 58.08%. This shows that deleting those low-quality pretraining examples or replacing them with newly collected examples will be a good strategy to improve the performance. Note that the influence score computation is fast. For example, on the CIFAR-10 dataset, the time for computing influence function with respect to *all* pretraining data is 230 seconds on a single Tesla V100 GPU, where 200 iterations of Conjugate Gradient for 2 IHVPs in (9), (10) and (11).

## 4.3 The Finetuning Task's Similarity to the Pretraining Task

In this experiment, we explore the relationship between influence function score and finetuning task similarity with the pretraining task. Specifically, we study whether the influence function score will increase in absolute value if the finetuning task is very similar to the pretraining task. To do this, we use the CIFAR-10 embedding obtained from a "bird vs. frog" classification and test its influence function scores on two finetuning tasks. The finetuning task A is exactly the same as the pretraining "bird vs. frog" classification, while the finetuning task B is a classification on two other classes ("automobile vs. deer"). All hyperparameters used in the two finetuning tasks are the same. In Figure B in the Appendix, for both tasks we plot the distribution of the influence function values with respect to each pretraining example. We sum up the influence score for all test examples. We can see that, the first finetuning task influence function has much larger absolute values than that of the second task. The average absolute value of task A influence function score is 0.055, much larger than that of task B, which is 0.025. This supports the argument that if pretraining task and finetuning task are similar, the pretraining data will have larger influence on the finetuning task performance.

## 4.4 Influence Function Score with Different Numbers of Finetuning Examples

We also study the relationship between the influence function scores and number of examples used in finetuning. In this experiment, we update the pretrained embedding in finetuning stage. We use the same pretraining and finetuning task as in Section 4.1. The results are presented in Figure C in the Appendix, model C is the model used in Section 4.1 while in model D we triple the number of finetuning examples as well as the number of finetuning steps. Figure C demonstrates the distribution of each pretraining examples' influence function score with the whole test set. The average absolute value of influence function score in model D is 0.15, much less than that of model C. This indicates that with more finetuning examples and more finetuning steps, the influence of pretraining data to the finetuning model's performance will decrease. This makes sense as if the finetuning data does not have sufficient information for training a good finetuning task, then pretraining data will have more impact on the finetuning task.

Table 1: Examples of test sentences and pretraining sentences with the largest and the smallest absolute influence function score values in our subset of pretraining data. The subset consists of 1000 random sentences from one-billion-word [3], which is used to pretrained ELMo embedding. Test examples are from a binary sentiment calssification task of Twitter.

| Test Sentence | Max absolute influence function value | Sentence in Pretrain | Min absolute influence function value | Sentence in Pretrain |
|---|---|---|---|---|
| *JebBush said he cut FL taxes by $19B. But that includes cuts in estate taxes mandated by federal law.* | 0.0841 | *Specifically , British Prime Minister Gordon Brown has recommended that security control in five provinces be handed over by the end of 2010.* | $4.5396 \times 10^{-6}$ | *One red shirt suffered a gunshot wound , most likely from a rubber bullet.* |
| *Creating jobs is our greatest moral purpose because they strengthen our families and communities.* | 0.0070 | *And Friday, the Commerce Department reports on durable goods orders and the University of Michigan releases its reading on consumer sentiment.* | $-1.3393 \times 10^{-7}$ | *Then there is the issue of why readers buy print publications , and whether the content they are buying can be consumed more easily or conveniently on the Internet .* |
| *The Kryptonian science council was more worried about climate change than these scary people.* | 0.0102 | *In addition the PTM also runs a vast network of mobile courts in the rest of the Fata areas, he said .* | $-1.999 \times 10^{-6}$ | *Until recently , such terror attacks inside Iraq could have coerced the village into sheltering Al Qaeda.* |

## 4.5 Quantitative Results on NLP Task

In this section we show the application of our proposed model on NLP task. In this experiment, the pretrain task is training ELMo [20] model on the one-billion-word (OBW) dataset [3] which contains 30 million sentences and 8 million unique words. The final pretrained ELMo model contains 93.6 million parameters. The finetune task is a binary sentiment classification task on the First GOP Debate Twitter Sentiment data[1] containing 16654 tweets about the early August GOP debate in Ohio. The finetune model uses original pretrained ELMo embedding and a feed-forward neural network with hidden size 64 to build the classifier. The embedding is fixed in the finetuning task. To show quantitative results, we randomly pick a test sentence from the finetune task, and sample a subset of 1000 sentences from one-billion-word dataset to check the influence of this test sentence to these data from the pretrain task. In Table 1 we show examples of test sentences and pretraining sentences with the largest and the smallest absolute influence function score values. Note that the computation time on this large-scale experiment experiment (the model contains 93.6 million parameters) is reasonable – each pretrain data point takes average of 0.94 second to compute influence score. For extremely large models and data sets, the computation can be further sped up by using parallel algorithm as each data point's influence computation is independent.

## 5 Conclusion

We introduce a multi-stage influence function for two multi-stage training setups: 1) the pretrained embedding is fixed during finetuning, and 2) the pretrained embedding is updated during finetuning. Our experimental results on CV and NLP tasks show strong correlation between the score of an example, computed from the proposed multi-stage influence function, and the true loss difference when the example is removed from the pretraining data. We believe our multi-stage influence function is a promising approach to connect the performance of a finetuned model with pretraining data. As future works, we will study our proposed method on other state-of-the-art ML models such as ResNet model and BERT model. Also it is interesting to investigate whether there are some pretraining examples which are task-agnostic with high/low influence scores, and removing/adding them can improve performance for various finetune tasks.

## Broader Impact

Multi-stage training has been used in many real applications but currently there is no data attribution method to explain how pretraining data influence the model's prediction in the end tasks. This paper provides a tool for doing this, which will be widely used in model debugging and fairness evaluation. For instance, our method can be used to investigate whether the bias of the end model is caused by a certain biased or malicious data in the pretraining stage. Although not mentioned in this paper, our method implicitly provides a way for data poisoning attack (similar to the origninal influence function paper). However, attacks will require full knowledge of training data and model, so we believe it is still impractical at the current stage.

## Footnotes

[1] https://www.kaggle.com/crowdflower/first-gop-debate-twitter-sentiment

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
