[Supplementary Material]

# A  Proof of Theorem 1

*Proof.* Since $\hat{\Theta}_\epsilon, \hat{U}_\epsilon, \hat{W}_\epsilon$ are optimal solutions, and thus satisfy the following optimality conditions:

$$0 = \frac{\partial}{\partial \Theta} F(\hat{W}_\epsilon, \hat{\Theta}_\epsilon) \tag{17}$$

$$0 = \frac{\partial}{\partial(W,U)} G(\hat{W}_\epsilon, \hat{U}_\epsilon) + \epsilon \frac{\partial}{\partial(W,U)} g(z, \hat{W}_\epsilon, \hat{U}_\epsilon), \tag{18}$$

where $\partial(W,U)$ means concatenate the $U$ and $W$ as $[W,U]$ and compute the gradient w.r.t $[W,U]$. We define the changes of parameters as $\Delta W_\epsilon = \hat{W}_\epsilon - \hat{W}$, $\Delta \Theta_\epsilon = \hat{\Theta}_\epsilon - \hat{\Theta}$, and $\Delta U_\epsilon = \hat{U}_\epsilon - \hat{U}$. Applying Taylor expansion to the rhs of (18) we get

$$0 \approx \frac{\partial}{\partial(W,U)} G(W^*, U^*) + \frac{\partial^2 G(W^*, U^*)}{\partial(W,U)^2} \begin{bmatrix} \Delta W_\epsilon \\ \Delta U_\epsilon \end{bmatrix} \tag{19}$$
$$+ \epsilon \frac{\partial g(z, W^*, U^*)}{\partial(W,U)} + \epsilon \frac{\partial^2 g(z, W^*, U^*)}{\partial(W,U)^2} \begin{bmatrix} \Delta W_\epsilon \\ \Delta U_\epsilon \end{bmatrix}$$

Since $W^*, U^*$ are optimal of unperturbed problem, $\frac{\partial}{\partial(W,U)} G(W^*, U^*) = 0$, and we have

$$\begin{bmatrix} \Delta W_\epsilon \\ \Delta U_\epsilon \end{bmatrix} \approx - \left( \frac{\partial^2 G(W^*, U^*)}{\partial(W,U)^2} + \epsilon \frac{\partial^2 g(z, W^*, U^*)}{\partial(W,U)^2} \right)^{-1} \tag{20}$$
$$\cdot (\frac{\partial g(z, W^*, U^*)}{\partial(W,U)}) \epsilon.$$

Since $\epsilon \to 0$, we have further approximation

$$\begin{bmatrix} \Delta W_\epsilon \\ \Delta U_\epsilon \end{bmatrix} \approx \left( \frac{\partial^2 G(W^*, U^*)}{\partial(W,U)^2} \right)^{-1} (\frac{\partial g(z, W^*, U^*)}{\partial(W,U)}) \epsilon. \tag{21}$$

Similarly, based on (17) and applying first order Taylor expansion to its rhs we have

$$0 \approx \lambda \frac{\partial F(W^*, \Theta^*)}{\partial \Theta} + \lambda \frac{\partial^2 F(W^*, \Theta^*)}{\partial \Theta \partial W} \cdot \Delta W_\epsilon + \lambda \frac{\partial^2 F(W^*, \Theta^*)}{\partial \Theta^2} \Delta \Theta_\epsilon. \tag{22}$$

Combining (21) and (22) we have

$$\Delta \Theta_\epsilon \approx (\lambda \frac{\partial^2 F(W^*, \Theta^*)}{\partial \Theta^2})^{-1} \cdot (\lambda \frac{\partial^2 F(W^*, \Theta^*)}{\partial \Theta \partial W}) \tag{23}$$
$$\cdot \left[ (\frac{\partial^2 G(W^*, U^*)}{\partial(W,U)^2})^{-1} (\frac{\partial g(z, W^*, U^*)}{\partial(W,U)}) \right]_W \epsilon$$

where $[\cdot]_W$ means taking the $W$ part of the vector. Therefore,

$$I_{z,W} := \frac{\partial \hat{W}_\epsilon}{\partial \epsilon} \Big|_{\epsilon=0} = - \left[ (\frac{\partial^2 G(W^*, U^*)}{\partial(W,U)^2})^{-1} (\frac{\partial g(z, W^*, U^*)}{\partial(W,U)}) \right]_W \tag{24}$$

$$I_{z,\Theta} := \frac{\partial \hat{\Theta}_\epsilon}{\partial \epsilon} \Big|_{\epsilon=0} = (\lambda \frac{\partial^2 F(W^*, \Theta^*)}{\partial \Theta^2})^{-1} \cdot (\lambda \frac{\partial^2 F(W^*, \Theta^*)}{\partial \Theta \partial W}) \tag{25}$$
$$\cdot \left[ (\frac{\partial^2 G(W^*, U^*)}{\partial(W,U)^2})^{-1} (\frac{\partial g(z, W^*, U^*)}{\partial(W,U)}) \right]_W.$$

$\square$

# B  Models and Hyperparameters for the Experiments in Sections 4.1, 4.2, 4.3 and 4.4

The model structures we used in Sections 4.1, 4.2, 4.3 and 4.4 are listed in Table A. As mentioned in the main text, for all models, CNN layers are used as embeddings and fully connected layers are task-specific. The number of neurons on the last fully connected layer is determined by the number of classes in the classification. There is no activation at the final output layer and all other activations are Tanh.

- For MNIST experiments in Section 4.1 on embedding fixed, we train a four-class classification (0, 1, 2, and 3) in pretraining. All examples in the original MNIST training set with with these four labels are used in pretraining. The finetuning task is to classify the rest six classes, and we subsample only 5000 examples to finetune. The pretrained embedding is fixed in finetuning. We run Adam optimizer in both pretraining and finetuning with a batch size of 512. The pretrained and finetuned models are trained to converge. When validating the influence function score, we remove an example from pretraining dataset. Then we re-run the pretraining and finetuning process with this leave-one-out pretraining dataset starting from the original models' weights. In this process, we only run 100 steps for pretraining and finetuning as the models converge. When computing the influence function scores, the damping term for the pretrained and finetuned model's Hessians are $1 \times 10^{-2}$ and $1 \times 10^{-8}$, respectively. We sample 1000 pretraining examples when computing the pretraind model's Hessian summation.

- For CIFAR experiments on embedding fixed, we train a two-class classification ("bird" vs "frog") in pretraining. All examples in the original CIFAR training set with with these four labels are used in pretraining. The finetuning task is to classify the rest eight classes, and we subsample only 5000 examples to finetune. The pretrained embedding is fixed in finetuning. We run Adam optimizer to train both pretrained and finetuned model with a batch size of 128. The pretrained and finetuned models are trained to converge. When validating the influence function score, we remove an example from pretraining dataset. Then we re-run the pretraining and finetuning process with this leave-one-out pretraining dataset starting from the original models' weights. In this process, we only run 6000 steps for pretraining and 3000 steps for finetuning. When computing the influence function scores, the damping term for the pretrained and finetuned model's Hessians are $1 \times 10^{-8}$ and $1 \times 10^{-6}$, respectively. Same hyperparameters are used in experiments in Sections 4.3 and 4.4. We also use these hyperparameters in with embedding unfix on CIFAR10's experiments, except that the pretrained embedding is updated in finetuning and the number of finetuning steps is reduced to 1000 in validation. The $\alpha$ constant in Equation 15 is chosen as 0.01. We sample 1000 pretraining examples when computing the pretrained model's Hessian summation.

| Dataset | MNIST | CIFAR |
|---|---|---|
| Embedding | CONV 32 5×5+1<br>MAX-POOL 2×2 +2<br>CONV 64 5×5+1<br>MAX-POOL 2×2 +2 | CONV 32 3×3+1<br>CONV 64 4×4+1<br>MAX-POOL 2×2 +2<br>CONV 128 2×2+1<br>MAX-POOL 2×2 +2<br>CONV 128 2×2+1<br>MAX-POOL 2×2 +2 |
| Task specific | FC <# classes> | FC 1500<br>FC <# classes> |

Table A: Model Architectures. "CONV $k\ w \times h + s$" represents a 2D convolutional layer with $k$ filters of size $w \times h$ using a stride of $s$ in both dimensions. "MAX-POOL $w \times h + s$" represents a 2D max pooling layer with kernel size $w \times h$ using a stride of $s$ in both dimensions. "FC n" = fully connected layer with $n$ outputs. All activation functions are Tanh and last fully connected layers do not have activation functions. The number of neurons on the last fully connected layer is determined by the number of classes in the task.

## C   Additional Experimental Results

To demonstrate the effectiveness of the expensive inverse Hessian computation in our formulation. We replace all inverse Hessians in (11) with identity matrices to compute the influence function score for the MNIST model as a comparison to Figure A . The results are shown in Figure 2(c) with a much smaller Pearson's r of 0.17. This result shows effectiveness of our proposed influence function.

In Figure B, we plot the distribution of the influence function values with respect to each pretraining example for two tasks described in Section 4.3. The finetuning task A is exactly the same as the pretraining "bird vs. frog" classification, while the finetuning task B is a classification on two other

classes ("automobile vs. deer"). We can see that, the first finetuning task influence function has much larger absolute values than that of the second task.

In Figure C, we plot the influence function score with different numbers of finetuning examples as introduced in Section 4.4. Model C is the model used in Section 4.1 and in model D we triple the number of finetuning examples as well as the number of finetuning steps.

Figure A: True loss difference vs. the influence function scores by our proposed method on MNIST. Pearson $r = 0.47$. Embedding is fixed in finetuning task.

Figure B: Two different finetuning task distribution of influence function scores. The pretrained embedding is fixed in finetuning. For both finetuning tasks, the pretrained model is the same, and is trained using "bird vs. frog" in CIFAR-10. For model A, finetuning task and pretraining task are the same. The average absolute values of influence function scores for models A and B are 0.055 and 0.025, respectively.

Figure C: Two different finetuning task distribution of influence function scores. The pretrained embedding is also updated in finetuning. The pretraining and finetuning tasks are the same as in Section 4.1. Model D's number of finetuning examples and finetuning steps are 3X of model C's. The average absolute values of influence function scores for Models C and D are 0.22 and 0.15, respectively.