[Reviews · NeurIPS 2020]

Review 1

Summary and Contributions: The paper proposes a novel method to estimate the influence score for multi-stage training, which is widely used in current state-of-the-art systems. The authors also empirically show the proposed method’s effectiveness with experiments.

Strengths: The proposed method is novel and technically sound. The problem the authors trying to address is important for many state-of-the-art deep models.

Weaknesses: The empirical studies in its current form are weak. The authors only conduct experiments on synthetic/simplified datasets, namely Cifar10 with two selected classes and MNIST with four classes. Also, for the NLP task, the authors only present results with a few selected samples. Such empirical study makes the paper quite weak. I think the paper would be significantly improved if the authros could apply the proposed method to real tasks to show its effectiveness. For example, research has shown that ImageNet contains some corrupted/incorrect labelled images and thus negatively impact some systems’ performance. I wonder if the proposed method can be used to improve the performance of those systems? *********************************after rebuttal*************************** Thank you for your feedback on my review; really appreciate that. After reading the other reviews and the rebuttal. This remains for me a marginal paper.

Correctness: The method sounds interesting and correct. But I think the empirical studies are quite weak.

Clarity: The paper is generally well written and easy to follow. Typo: line 29: “. From a robust…”

Relation to Prior Work: Yes; to the best of my knowledge, the prior work is well presented and discussed.

Reproducibility: Yes

Additional Feedback: Can the proposed method be used to guide the pre-training processes to improve the current state-of-the-art systems for some real tasks using BERT pre-training or ImageNet pre-training?


Review 2

Summary and Contributions: This paper addresses a longstanding and difficult problem: how can one trace backward from a final result, through various layers of transformation, to determine the culprits in the initial training set for a poor machine learning model? This paper proposes a multi-stage influence function ‘propagation’ score to trace eventual labels back to their sources. It is an extension to multiple layers of (I presume) the same authors’ single-stage model from 2017.

Strengths: The paper is clear and logically organized. I did not have time to check the math in Section 3, but the experiments in Section 4 are well-crafted and executed and the various kinds of results seem convincing, though lacking a higher-level discussion of are properly reported.

Weaknesses: What is somewhat lacking though is a higher-level discussion of what one could/should expect in principle to be actually useful; for example, let’s say you can identify a few problematic samples in a training set, is it sufficient to simply delete them? How much damage would you do thereby to predictions for other tasks in the future? That is, how ‘stable’ is the initial training data to perturbations, and the subsequent one?

Correctness: I did not do a thorough check but seems ok on the surface.

Clarity: Yes

Relation to Prior Work: Yes

Reproducibility: Yes

Additional Feedback:


Review 3

Summary and Contributions: Influence function is used to identify the training examples which contribute most to the prediction of the model. This paper studies multi-stage influence functions, which means that the model is trained by more than one stages, e.g., firstly pre-trained on an auxiliary task and then finetuned on the target task. The paper studies the multi-stage Influence functions under two different fine-tuning settings: (1) fine-tuning all the parameters, and (2) fine-tuning partial parameters. Experiments are conducted to validate the proposed method.

Strengths: 1. The authors introduce an interesting and understudied problem. The paper is overall clearly written and easy to follow, although I have no prior experience with influence functions. 2. Two widely used fine-tuning methods, fine-tuning all the parameters and fine-tuning partial parameters, are discussed to devise the influence functions. Experiments show somewhat positive results to validate the proposed method.

Weaknesses: 1. My main concern is that the authors did not compare the proposed method with any latent baselines. For example, we can also use to the uncertainty (the loss value or the entropy of the predicted distribution) of the model as an indicator to identify those problematic examples in the pre-training data. As the proposed method in this paper does not show very impressive results in the experiments (the Pearson’s correlation is only 0.4~0.6 in Fig. 1), it may not outperform this simple baseline. 2. In the experiments, the transfer tasks come too artificially. “At the pretraining stage, we train the models with examples from two classes (“bird" vs. “frog") for CIFAR-10 and four classes (0, 1, 2, and 3) for MNIST”. The transfer tasks in these settings may be too easy. In addition, the experiments are limited to small datasets like MNIST and CIFAR-10. I wonder how the proposed method scale to larger datasets or more challenging transfer tasks.

Correctness: Yes

Clarity: Yes

Relation to Prior Work: Yes

Reproducibility: Yes

Additional Feedback:


Review 4

Summary and Contributions: This paper addresses the problem of estimating the influence of specific training data points on the predictions of a model. Specifically, the work considers multi-stage training where a model is first pre-trained and then finetuned to a task. The paper considers two cases --- first, when the pretrained parameters are kept fixed; and second, when the pretrained parameters are also updated during finetuning --- and derives efficient algorithms to compute the influence scores under these settings. The experiments showed that the proposed approach can approximate the true influence of pre-training data points in downstream models and can be used to clean the training data and thus avoid negative transfer issues.

Strengths: Deep learning models induced by multi-stage training are increasingly becoming the norm in ML research and applications and yet the underlying mechanisms the govern this process are still poorly understood. This paper is a step in the direction of a creating more systematic and controlled methodologies for multi-stage training.

Weaknesses: None == Updated == I agree that the empirical is somewhat weak in that it does not reflect the difficulty and scale of modern ML/DL tasks. However, I understand that it would be very computationally demanding to make those experiments.

Correctness: The proposed approach is sound and the experimental results are convincing.

Clarity: The paper is well written and organized

Relation to Prior Work: Yes

Reproducibility: Yes

Additional Feedback: typo in line 84: perspective instead of prospective

[Author Response · NeurIPS 2020]

**We thank all four reviewers for their great reviews. We provide our feedback for each reviewer as follows.**

**General concerns on experiment designs from R1 and R3:**

The main goal of the empirical study in our paper is 1) to validate the proposed influence function is a good approximation to the ground truth influences and 2) to show the scalability. To validate the correctness, one needs to **train the pretraining model from scratch with each training sample removed** and retrain the finetuning task to get the ground truth influence. Due to the difficulty of evaluation, we can only use small scale datasets (e.g. MNIST and CIFAR) to generate figures like Figure 1 (a-c) in our paper to show the correctness. In fact, small datasets such as MNIST and CIFAR are standard datasets in influence function or data cleansing literature. In related works such as Koh and Liang, ICML'17, Koh et al. NeurIPS'19, and Hara et al. NeurIPS'19, the researchers also evaluated their methods on these datasets or similar scale datasets.

In this work, in addition to validate the correctness, we further demonstrate that scalability is not a problem for our algorithm – the computation can scale to a large and complex NLP task: Elmo model+sentiment analysis task. The Elmo model has 93.6 million parameters and is trained using 30 million sentences; however it's almost impossible to plot Figure 1 for Elmo to validate the correctness since to do so we need to train the Elmo model from scratch with each sample removed which takes months.

We thank the reviewer R1 for pointing out a potential large scale experiment on ImageNet with interesting applications, but we were not able to finish that in the limited rebuttal time as for this task we need to remove one sample in ImageNet at a time for all the images, and train a ResNet model from scratch to validate the correctness. We will add that in a revised version.

**R1.**   We thank the reviewer for the insightful review. please see the above 'General concerns on experiment designs' section for details about our empirical study. We hope this can address your concern.

**R2.**   We thank the reviewer for the positive comments.

High-level discussion:  We will add a section in the revised version on higher-level discussion about. From the experimental results in Section 4.1 we can see that, removing the pretrain examples with high positive influence function values will decrease the model's total loss values on the test set and thus improve model's performance in testing. Hence, deleting those low-quality pretraining examples or replacing them with newly collected examples would be a good strategy.

Generalization to other finetune tasks:  Our proposed multi-stage influence function is finetune task specific (see the influence score computation in Eq(11)). Please refer to Section 4.2 for finetune tasks' accuracy after removing low quality pretrain examples identified by our influence function. However, if we use one finetune task's influence function to clean pretrain data for another finetune task, it is hard to tell the performance. We design a new experiment on that: similar to the settings in Section 4.1, we use a 4-class classification (0, 1, 2, and 3) task from MNIST as pretrain task and a 3-class classification (4, 5, 6) task as finetune task to calculate influence function. Then we remove the pretrain data and go through the same process to get the finetune loss difference of another classification task (7, 8, 9). The Pearson $r$ value of this task is only 0.13. This experiment illustrates that the low quality examples identified by influence function for one finetune task may not be low quality for another finetune task. While it is still interesting as future work to see whether that are some pretrain examples that are task-agnostic with high/low influence scores and removing/adding them can improve performance for various finetune tasks.

**R3.**   We thank the reviewer for the insightful review.

Latent baseline:  We thank the reviewer for proposing this baseline scenario. We implemented the reviewer's idea and use the pretrain loss value as the score and followed the procedures in Section 4.1 to evaluate its effectiveness. The Pearson r values are 0.09 for CIFAR and 0.10 for MNIST, which are much smaller than our proposed method. We will add this baseline in the revised version. We think that the main reasons are that: (1) Training examples with large training loss or large entropy values of the predicted distribution are not necessarily low quality examples. For example, in a binary soft SVM, samples in between two margins are support vectors that with high uncertainty, but it is uncertain whether the impact is positive or negative. (2) Even if we can identify low quality data in the pretrain task, because pretrain and finetune are different tasks, it is possible that those examples are actually helpful if we want the embedding to perform better in the finetune task. Please see our response to R2's "Generalization to other finetune tasks" for a relevant experiment we added.

Larger datasets:  please see the above 'General concerns on experiment designs' section for details about our empirical study. We hope this can address your concern.

[Meta-Review · NeurIPS 2020]

Experiments are somewhat limited and the method is computationally is expensive, but the problem definition is novel and sufficiently interesting to justify studying this problem.